# A Deep CNN Transformer Hybrid Model for Skin Lesion Classification of Dermoscopic Images Using Focal Loss

**DOI:** 10.3390/diagnostics13010072

**Published:** 2022-12-27

**Authors:** Yali Nie, Paolo Sommella, Marco Carratù, Mattias O’Nils, Jan Lundgren

**Affiliations:** 1Department of Electronics Design, Mid Sweden University, 85170 Sundsvall, Sweden; 2Department of Industrial Engineering, University of Salerno, 84084 Fisciano, SA, Italy

**Keywords:** deep learning, skin lesion, hybrid model, focal loss

## Abstract

Skin cancers are the most cancers diagnosed worldwide, with an estimated > 1.5 million new cases in 2020. Use of computer-aided diagnosis (CAD) systems for early detection and classification of skin lesions helps reduce skin cancer mortality rates. Inspired by the success of the transformer network in natural language processing (NLP) and the deep convolutional neural network (DCNN) in computer vision, we propose an end-to-end CNN transformer hybrid model with a focal loss (FL) function to classify skin lesion images. First, the CNN extracts low-level, local feature maps from the dermoscopic images. In the second stage, the vision transformer (ViT) globally models these features, then extracts abstract and high-level semantic information, and finally sends this to the multi-layer perceptron (MLP) head for classification. Based on an evaluation of three different loss functions, the FL-based algorithm is aimed to improve the extreme class imbalance that exists in the International Skin Imaging Collaboration (ISIC) 2018 dataset. The experimental analysis demonstrates that impressive results of skin lesion classification are achieved by employing the hybrid model and FL strategy, which shows significantly high performance and outperforms the existing work.

## 1. Introduction

According to the International Agency of Research on Cancer (IARC) report, in 2020 an estimated 325,000 new cases of melanoma were diagnosed worldwide and about 57,000 people died from the disease. Scientists from IARC predict that from 2020 to 2040 the number of new cases of cutaneous melanoma will increase by >50%, to >500,000 per year, and the number of deaths caused by melanoma will increase by over two-thirds, to almost 100,000 per year [1]. The cancer generally develops as a result of exposure to ultraviolet (UV) rays from the sun, which harms the deoxyribonucleic acid (DNA) of skin cells [2]. The gold standard for a diagnosis of invasive melanoma is the examination of histopathological whole slide skin biopsy images by an experienced dermatopathologist [3]. The cost per melanoma detected has been estimated at USD 32,594, and 30% of costs coming from biopsy [4]. To become a licensed dermatologist requires many years of education and training. On average, it takes at least 12 years of education and training after high school [5]. Likewise, training an inexpert dermatologist is also time-consuming. Dermatologists achieve only about 75% accuracy when diagnosing melanoma [6]. However, to improve the diagnosing rate of malignancy, a computer-aided diagnosis (CAD) system can be used to automatically classify skin lesions in dermoscopic images.

As compared with the binary class classification problem, multiclass classification is more complex due to the similarity between different types of skin lesions [7]. The International Skin Imaging Collaboration (ISIC) 2018 dataset was published by the ISIC as a large-scale dataset of dermoscopy images [8]. Its Task 3 concerns lesion classification, and contains a total 10,015 labeled dermoscopic images. It classifies the dermoscopic images into one of the following categories: melanoma (MEL), melanocytic nevus (NV), basal cell carcinoma (BCC), actinic keratosis (AKIEC), benign keratosis lesion (BKL), dermatofibroma (DF), and vascular lesion (VASC). The ISIC 2018 dataset includes seven classes, and there is a high imbalance between the different classes, as illustrated in Figure 1. Especially, there are around 58 times more samples in the NV class than there are DF class samples. This will degrade the overall performance of a network when it is biased towards classes with the greatest number of examples. This paper uses the public dataset of dermoscopic images provided by ISIC 2018.

In this work, our main contributions can be summarized as: We show that significant performance improvements can be achieved by proposing a deep CNN transformer hybrid framework. To the best of our knowledge, this is the first network to combine Transformers with CNN and improve the classification of skin lesions. In addition, we apply three different loss functions to demonstrate improvements in the imbalanced ISIC dataset of 2018.

The remainder of this paper is structured as follows: Section 2 presents related works. Section 3 describes the architecture of our hybrid CNN transformer model including three different loss functions to reduce the data imbalance problem and enhance the performance of the model. Section 4 describes the simulation environment and presents the performance metrics, experimental results, and a discussion. It also provides a performance comparison of the proposed solution with state-of-the-art methods. Finally, Section 5 concludes the paper and advances future research directions.

## 2. Related Works

Previously, handcrafted feature-based techniques were introduced for skin lesion classification [9,10,11]. However, these conventional machine-learning approaches did not achieve good results due to variations in the shape, color, and size of skin cancer.

In recent years, deep learning has been used for analyzing different medical images [7,12,13,14,15,16,17,18,19,20] and shows excellent performance. Different medical image categories include X-ray, magnetic resonance imaging (MRI), computed tomography (CT), dermoscopic and so on. Compared with traditional methods, techniques based on deep convolutional neural network (DCNN) can directly learn meaningful features from skin lesion datasets and have already achieved significantly improved performance [7,16,17,18,19,20]. For example, Kumar et al. [20] have claimed that using deep learning exhibits higher performance than all the other conventional models, such as random forest (RF), and support vector machine(SVM). Esteva et al. [17] utilized Inception-v3 CNN [21] architecture, which achieves performance on par with all tested experts, demonstrating an artificial intelligence (AI) capable of classifying skin cancer with a level of competence comparable to that of dermatologists. Pacheco et al. [22] analyzed the out-of-distribution detection in skin cancer classification of four competitive CNN models: DenseNet-121 [23], MobileNet-v2 [24], ResNet-50 [25], and VGGNet-16 [26], showing that the Mahalanobis and Gram Matrix-based methods achieve competitive performance. Table 1 summarized the findings and limitations of the related works. From this table, we can see that the biggest disadvantage of traditional handcrafted methods is that they can only train small datasets. The use of deep learning methods can solve this problem. For example, Esteva et al. [17] used 129,450 images to train the model. In our paper, we mainly study and discuss deep-learning methods for skin lesion classification.

Many recent works have striven to design more effective architecture for vision. The transformer was initially developed by Google [27] for the natural language processing (NLP) field and has had much success in this field. The previous recurrent neural network (RNN) model had limited memory and no way to parallelize the processing, in contrast to the transformer. Vision transformers (ViTs) have been shown by Steiner et al. to attain highly competitive performance for a wide range of vision applications [28]. Theirs was the first study to successfully train a transformer encoder on ImageNet, attaining very good results compared with familiar convolutional architectures. Tolstikhin et al. [29] presented a multi-layer perceptron (MLP) mixer, an architecture based exclusively on MLPs.

Moreover, some researchers [7,30,31,32,33] propose to combine different features, which can have an impressive impact on the classification performance. In the cited works, fusion was done by concatenation, or majority vote, using pre-computed representations in a low resolution. Afza et al. [7] described a method of five primary steps using a hybrid whale optimization algorithm and entropy and mutual information. Zhou et al. [34] proposed an approach consisting of self-attention blocks and guided-attention blocks to evaluate the Human Against Machine with 10,000 training images (HAM10000) dataset, with impressive performance. He et al. [35] employed the full transformer network, which is a hierarchical transformer computing feature using a spatial pyramid transformer.

The transformer is the first method that entirely depends on the self-attention mechanism to explore long-range dependency in NLP. The transformer layer comprises a multi-head attention (MHA) layer and an MLP. A skip connection and layer normalization are applied in the MHA and the MLP layers. On the other hand, the ViT is composed purely of transformer layers, which have been proposed for image recognition. It has been demonstrated that the transformer can achieve state-of-the-art performance [36].

Although CNNs have proved their effectiveness in skin lesion classification, they fail to effectively capture contextual information because of the intrinsic locality of the convolution operator [27,36]. Long-range dependencies are prerequisites for forming contextual information, and such information is crucial for classification tasks. So far, many solutions have been proposed to improve CNNs. Gessert et al. [30] relied on ensemble models for patient metadata. Dosovitskiy et al. [36] reported that sequences of image patches can perform very well on image classification tasks and that ViT attains excellent results. CNN-based methods attempt to explore global features by gradually expanding the receptive field while ignoring long-range contextual information. Vision transformers can extract contextual features, but the learning ability regarding local information is limited, and there is a significant computational complexity simultaneously [37]. This paper was inspired by this challenge, and reports on applying a hybrid CNN transformer model to skin lesion classification.

**Table 1 diagnostics-13-00072-t001:** Summary of the selected related works.

	Paper	Study	Year	Proposed	Finding	Limitation
Phase						
Handcrafted		[9]	2014	Proposed an automatic diagnosis of melanoma based on a 7-point checklist	It used traditional machine learning methods, first boundary detection and then feature extraction, final make a classification	Very small dataset, only 300 images; Private dataset, cannot be reproduced
		[10]	2007	Used the relative color histogram analysis technique to evaluate skin lesion discrimination in dermoscopic images	Traditional handcrafted method focus on color histogram analysis	Small dataset with two classes, 113 malignant and 113 benign
		[11]	2013	Proposes a hierarchical classification system with the K-Nearest Neighbors	Hierarchical structure decomposed the skin lesion classification into a set of simpler problems; Color and texture information from images acquired by a standard camera (non-dermoscopy)	Private dataset; Small dataset, 960 images; Only two hierarchical levels
Deep-learning		[7]	2022	Applied hybrid deep features selection on skin lesion classification	using hybrid whale optimization select feature; contrast enhancement	HAM10000 and ISIC2018 are same training dataset; There is no clear indication about test dataset, and there is no standard benchmark
		[16]	2015	Combined deep learning, sparse coding, and SVM learning algorithms	Using sparse coding as unsupervised feature learning	Only using three metrics: accuracy, sensitivity, specificity; Only three classes: melanoma, atypical and benign
		[17]	2017	Used Google’s Inception v3 CNN architecture to classify skin cancer	The CNN achieves classifying skin cancer with a level of competence comparable to dermatologists; Trained a CNN using a dataset of 129,450 clinical images comprising 2032 different diseases	The dataset is huge but not all available since some come from the Stanford Hospital
		[18]	2022	Proposed a new residual deep convolutional neural network for skin lesions	Trained and tested using six skin cancer datasets, PH2, DermIS and Quest, MED-NODE, ISIC2016, ISIC2017, and ISIC2018	Innovation is not obvious, a deep neural architecture based on residual learning
		[19]	2022	Proposed deep metric attention learning CNN for skin lesion classification	With a triplet-based network; using a mixed attention mechanism; hybrid loss function	Only experimented on small datasets (ISIC 2016, ISIC 2017, and PH2), did not use larger datasets, such as ISIC2018
		[20]	2021	Proposed fusion-based Deep learning methodology(RF, SVM, ANN)	Using sum rule fusion method; Metric, Mean Square Error, Peak Signal to Noise ratio for assessing the quality of the pre-processing strategy	Insufficient comparison test
		[33]	2020	Proposed global-par CNN model with data-transformed ensemble learning and evaluated on the ISIC 2016 and ISIC 2017	Three fusion strategies: averaging predictions, SVM stacking, and a weighted ensemble of predictions; Data-transformed ensemble learning	Singe modality of dermoscopy image cannot provide enough information to classify the melanoma; future(dermoscopy image, clinical image, and patient’s metadata)
		[34]	2021	Propose a Mutual Attention Transformer(MAT) for skin lesion diagnosis	Fine-grained features from the two modalities (image and text); The MAT consists of self-attention blocks and guided-attention blocks	To deal with the problem of data imbalance, simply use data augmentation
		[38]	2018	Proposed ensembled deep convolutional neural networks on skin lesion classification	Fuse the outputs of the classification layers of four different deep neural network architectures; Different fusion-based methods; Have a unified benchmark	Small dataset with 2000 images; Not experimented with larger databases
		[39]	2019	Proposed Bayesian deep networks for skin lesion diagnosis	Bayesian deep networks can boost the diagnostic performance without incurring additional parameters or heavy computation	Only accuracy is used, and several other standard classification criteria are missing. No comparison with other researchers’ experimental results

Dataset imbalance can cause two problems [40]: Firstly, the training becomes ineffective, as most observations are easy samples (normal samples) that provide no learning benefit to the model. Secondly, normal samples can dominate the training and cause the classifier to favor classes with a large number of labeled samples. One common solution to address such a class imbalance problem is augmenting small classes with more data through replication or transformation operations(data augmentation). Recently, Lin et al. [41] developed the focal loss (FL) strategy, by defining the class weight factor as a function to improve the performances of detectors. This loss function can make the model applicable to minority classes.

## 3. Proposed Method

Convolutional neural networks may be indirectly limited [31] when trained with highly variable and distinctive image datasets with limited samples, such as dermoscopic image datasets. As shown in Figure 2, we utilized the hybrid neural network to build an automatic system to classify different class skin lesions. This is quite common in medical diagnostics as positive cases are often in the minority compared with negative cases [38]. To minimize these effects, we have also included cross-entropy (CE), weighted cross-entropy (WCE), and FL function to enhance the underrepresented categories. The following subsections will explain in detail.

### 3.1. Architecture of the Proposed Hybrid Model

The hybrid model combines traditional CNN feature extraction with vision transformation. The ViT component consists of three main parts: an embedding layer, a transformer encoder, and an MLP head. A more detailed structure is given in Figure 2. Being widely used, CNNs are usually composed of convolutional layers, pooling layers, fully connected (FC) layers, and so on. Benefiting from layer-by-layer structures, they allow abstract features to be gradually extracted from stacked layers. Various CNN models have been proposed in the past few years, ResNet being one of the most classic frameworks. (In the present study, ResNet-50 was selected for CNN feature extraction. The CNN model applied the first ten layers of Resnet-50 as the feature extractor, as shown in Figure 2b. Input dermoscopic images are processed by ResNet-50, which consists of a convolutional layer (7 × 7 kernel), a max pooling layer, and a series of residual blocks, followed by the patch embedding layer. (In this study, the kernel size and stride of the convolutional layer in patch embedding have become one, and this layer was only used for adjusting the channel.) The latter part is vision transformation. This needs to add the class token as well as position embedding before entering the transformer encoder. The class token and position embedding are trainable parameters. The transformer encoder is a network consisting of repeated stacking of encoder block L times. It mainly consists of the following parts: layer norm, multi-head attention, and MLP block, as shown on the right side of Figure 2. Multi-layer perceptron is composed of an FC layer, a Gaussian error linear unit (GELU) activation function, and a dropout layer.

Different from traditional attention [42,43,44], the adopted MHA feature maps a set of key-value (K,Q) pairs and a set of queries (*Vs*) to output vectors; and the dimensions of the three learnable matrices are $d_{k}$, $d_{k}$, and $d_{v}$, respectively. For each element in the input sequence *R*, the values of Q,K, and *V* are calculated by multiplying *R* by three learned matrices, $P_{QKV}\in R^{D\times 3d_{k}}$.
(1)$$\left[Q,K,V\right]=RP_{QKV}$$

As a core module of the ViT, the MHA feature allows the model to jointly attend to information from different representation subspaces at other positions. As shown in Figure 2d, the MHA calculates the scaled dot-product attention, to obtain the corresponding head. Before performing the scaled dot-product attention calculation, it performs three linear projections to transform the Qs, Ks, and Vs to more discriminated representations, respectively. Then, each head is concatenated and fed into another linear projection to obtain the final outputs of the MHA. Dot products are computed, and the Qs with all Ks are coupled with a scaling factor $\left(1/{\left(d_{k}\right)}^{1/2}\right)$, and then calculated using the softmax function on the Vs to form attention weights. Thus, the generated attention weights implying the key part of image patches are defined as follows:(2)$$Attention\left(V,K,Q\right)\;=\;Softmax\left(\frac{QK^{T}}{\sqrt{d_{k}}}\right)V$$

To reduce sequence computation, it is beneficial to map the Qs, Ks, and Vs*h* times with different linear projections and then perform Equation (Equation 2) in parallel. Subsequently, formed *h* vectors are concatenated via the concatenation layer, and each vector is called a “head”. After concatenation, the MHA results can be obtained as follows:(3)$$MHA\left(V,K,Q\right)\;=\;Concat\left(head_{1},\cdots ,head_{h}\right)W^{o},$$
where $head_{i}\;=\;Attention\left(QW_{i}^{Q},KW_{i}^{K},VW_{i}^{V}\right)$, $W^{o}\in R^{h\times d_{v}\times D}$ means learnable weights of a feedforward layer and Concat(·) denotes the operation of concatenation.

When the output of the MHA module is determined, the corresponding results are further processed by the MLP module illustrated in Figure 2e. In the MLP block, two FC layers with the GELU are stacked to produce the output of the single layer in the transformer encoder. The equation for the GELU is as follows:(4)$$GELU\left(x\right)\;=\;x\phi \left(x\right)=x\cdot \frac{1}{2}\left[1+erf\left(x/\sqrt{2}\right)\right],$$
where ϕ(x) represents the standard Gaussian cumulative distribution, and
(5)$$erf\left(x\right)=\left(2/\sqrt{\pi }\right)\int_{0}^{x}e^{-\eta^{2}}d\eta$$

Furthermore, the layer norm(LN) in the transformer encoder is calculated as follows:(6)$$\mu^{f}=\frac{1}{H}\sum_{i=1}^{H}a_{i}^{f}$$
(7)$$\sigma^{f}=\sqrt{\frac{1}{H}\sum_{i=1}^{H}{\left(a_{i}^{f}-\mu^{f}\right)}^{2},}$$
where *H* represents the number of hidden units of a layer, $a_{i}^{f}$ denotes the summed inputs of the *i*th hidden unit in the *f*th layer, and all hidden units in a layer share the same normalization terms μ and σ. Thereafter, the normalized hidden unit ${\hat{a}}^{f}$ can be generated via
(8)$${\hat{a}}^{f}=\frac{a^{f}-\mu^{f}}{\sqrt{{\left(\sigma^{f}\right)}^{2}+\varepsilon }},$$
where ε is a constant for numerical stability.

### 3.2. Loss Function

In this section, we first discuss the common loss functions: CE, WCE loss and FL. We then introduce our proposed hybrid model, which is a CNN mixed with a ViT.

#### 3.2.1. Cross-Entropy Loss

Cross-entropy loss [45] is often used in multi-class classification. Because CE involves calculating the probability of each category, it is almost always used with the softmax [46] (or the sigmoid) function come together. In a neural network, we usually use the softmax layer to obtain a multi-class predicted probability distribution, and then we apply CE to calculate a score that summarizes the average difference between the predicted and the actual probability distributions for all classes in the problem. The score is minimized; a perfect CE value is 0.

Figure 3 shows the process of gaining loss and learning, and the model’s prediction process. The final layer of the neural network receives the score from each category scores. The score is passed into the softmax function to obtain the probability output. The class probability output predicted by the model is the same as that of the real class one hot form to compute the CE loss. Ultimately, there are only two classes that the model needs to predict at the end in binary classification. For each class, the probability of the prediction is *p* and 1−p. The formula of CE loss function [47] is expressed as follows::(9)$$L_{CE}\;=\;-\sum_{i}^{M}y_{ic}log\left(p_{ic}\right)$$
M = number of categories (M>2).$y_{ic}$ = symbolic function ( 0 or 1 ), if the true class of the *c* sample *i* is equaled 1, or otherwise takes 0.$p_{ic}$ = the predicted probability that the observed sample *i* belongs to the class *c*.$z_{i}$ = the output value of the *i*th node,where $p_{ic}$ is obtained by applying the softmax function, which ensures that the output of the function is a value between 0 and 1.
(10)$$p_{ic}\;=\;softmax\left(z_{i}\right)\;=\;\frac{e^{z_{i}}}{\sum_{M}e^{z_{M}}}$$

#### 3.2.2. Weighted Cross-Entropy Loss

Since skin lesion classification is treated equally in the form of CE loss (Equation (Equation 9)), some categories with few samples do not make a significant contribution to the total loss and thus tend to be ignored during training. An effective way to remedy this problem is to assign suitable weights to each object class as represented in Equation (Equation 11) below. The idea with WCE is to use a coefficient to describe the impact of sample loss [48]: For a small number of samples, this will strengthen the contribution to loss, while for a large number of samples, there will be reduced loss. This is only a small change from binary CE, that is, a $w_{i}$ coefficient is added to the discrimination of positive samples to control the balancing severity. $w_{i}$ is calculated in advance based on he dataset. The defined weight vector $w_{i}\in R^{M}$ with elements $w_{i}>0$ is defined over the range of class labels i∈{1,2,⋯,M}. The more examples in the training data, the smaller the weight of the loss. The method is to normalize the weights proportionally to the reverse of the initial weights.
(11)$$L_{WCE}\;=\;-\sum_{i}^{M}w_{i}y_{ic}log\left(p_{ic}\right)$$

#### 3.2.3. Focal Loss

To further enhance the model’s performance, we employed a loss function called “focal loss” [41]. This is a special type of categorical loss that seeks a solution to data imbalance. It considers the contribution of each sample to the loss based on the classification error. Using this function, the loss decreases when the model correctly classifies a sample. Focal loss focuses training on a small subset of hard examples and avoids a large number of easy negatives from overwhelming the classifier during training. This idea is mathematically expressed in Equation (Equation 12). To increase the weight of difficult samples and reduce the weight of simple samples, an adjustment factor is added on the basis of the CE loss function. Obviously, with FL, easily classified samples are diminished while hard-to-classify samples have greater loss values, which makes the model pay more attention to these samples, and as a result, improves the precision for them. This approach solves the class imbalance problem by making the loss indirectly focus on challenging classes.
(12)$$L_{FL}\;=\;-\alpha \sum_{i}^{M}{\left(1-p_{ic}\right)}^{\gamma }y_{ic}log\left(p_{ic}\right)$$

In Equation (Equation 12), α is used to deal with the problem of class imbalance, whereas γ is used to solve the problem of unbalanced difficult and easy samples. They are two adjustable parameters.

## 4. Experimental Results and Discussion

To evaluate the effectiveness of the approach, we compared the performance of the proposed deep hybrid model with that of the original Resnet-50 model. We also conducted experiments to evaluate the three different loss functions in skin cancer classification. We compared the results with recent studies to examine the performance of the proposed method.

### 4.1. Implementation Details

We conducted six models in the experiment, CNNce, CNNwce, CNNfc, CNNViTce, CNNViTwce, and CNNViTfc. We split the ISIC 2018 dataset into a training set, a validation set and a testing set with a split ratio of 0.7, 0.1, and 0.2. The experiment in this paper used the Pytorch framework and Python language to implement the proposed approach.

Parameter settings have a significant impact on experimental results. In this paper, the most suitable training strategy and hyperparameters were determined in numerous experiments. The training settings were as follows: We used Cosine Annealing as the learning rate schedule, and the initial learning rate was 0.002. We further used a CE function with label smoothing, with the label smoothing factor set to 0.1. Adam optimization algorithm was applied as an optimizer. The weight decay coefficient was 0.001, the batch size was 32, and all models were trained until convergence was reached. Batch size and learning rate were based on the graphics processing unit (GPU) memory requirements of each architecture. We evaluated every five epochs and saved the model that achieved the best mean class recall. Training was performed on NVIDIA GTX 2080TI graphics cards. The parameter values were α = 0.25 and γ = 2 in the FL function, demonstrated in the literature to be optimum values leading to the best performance [41].

Before feeding the dermoscopic images to the CNN, we performed extensive data augmentation. We used random flipping, random rotation, brightness and contrast changes, random scaling, and random cropping. Furthermore, we used color jitter and random affine transformation.

### 4.2. Evaluation Metrics

To evaluate the effects using different skin cancer classification models, we applied multiple evaluation criteria including accuracy, precision, recall (=sensitivity), and F1 score, as shown in Equations (Equation 13)–(Equation 16). Accuracy can clearly be used to judge the performance of our model, but there is a serious flaw: When the sample proportion of different categories is unbalanced, the category with the largest proportion will often become the most important factor affecting the accuracy, and this case the accuracy is not very good. This reflects the overall situation of the model. The confusion matrix can clearly display the number of correctly classified samples for each class and the details for each misclassified class. However, it is not easy to evaluate the performance of various classification models based on the confusion matrix. Therefore, a variety of classification accuracy indicators are employed, among which overall accuracy (OA), macro average (macro avg), and area under the receiver operating characteristic (ROC) curve (AUC) are the most widely used.
(13)$$Accuracy\;=\;\frac{TP+TN}{TP+TN+FP+FN}$$
(14)$$Precision\;=\;\frac{TP}{TP+FP}$$
(15)$$Recall\;=\;\frac{TP}{TP+FN}$$
(16)$$F1-score\;=\;\frac{2\times (Precision\times Recall)}{Precision+Recall}$$
(17)$$AUC\;=\;\frac{\sum_{i\in positiveClass}rank_{i}-\frac{M(1+M)}{2}}{M\times N}$$

TP = True positive.FP = False positive.FN = False negative.TN = True negative.$rank_{i}$ = the serial number of the *i*th sample. (Probability scores are ranked from small to large, in rank position).M,N = the number of positive and negative samples, respectively.$\sum_{i\in positiveClass}$ = only using the serial numbers of the positive sample.

“Macro average” essentially refers to the arithmetic mean of each statistical indicator value for all categories, so the simple mean ignores the situation that there may be a great imbalance in the distribution of samples. To solve the issue that the sample imbalance is not considered in Macro, when calculating Precision and Recall, the Precision and Recall of each category is multiplied by the proportion of the category in the total sample to be summed up(weighted average). “Micro average” computes a global average by counting the sums of the true positives (TPs), false negatives (FNs), and false positives (FPs).

Area under the ROC curve [49], is a two-dimensional plot of the TP rate. The value of this area is between 0 and 1, which can intuitively evaluate the quality of the classifier. The larger the AUC value, the better the classifier effect a perfect model has an AUC close to one.

### 4.3. Results, and Discussion

We trained the six different models and observed the rate of convergence of the model with 1000 epochs. In addition, we began saving the model providing the highest mean class recall value. To further illustrate the effectiveness of the proposed method, we present the confusion matrices of all models in Figure 4.

To further intuitively evaluate the confusion degree and different class classification errors, we provided six confusion matrices on the ISIC 2018 dataset by using the six proposed models. Figure 4 presents the confusion matrices’ experimental results, which indicate an improvement in correct classification of the classes, such as NV, AKIEC, BKL, DF, and VASC. For example, the correct classification improved from 1020 to 1239 for the NV class and from 133 to 168 for the BKL class, as shown in Figure 4a,b. Even for the BCC class, the number of correctly classified images slightly increased, from 83 to 84. This was due to the hybrid model, which improved the classification performance. However, the model also misclassified some MEL lesions from the normal class as belonging to other classes (note that the number of correctly classified MEL lesions decreased from 175 to 155). The correct prediction of each related diagnostic category is given on the main diagonal of each matrix. It can be concluded that ViT models did a poor job on MEL classification. In the same manner, FL models did much better on the class of AKIEC than CE loss. Overall, hybrid model ViT with FL has achieved the best results. Specifically, the OA was improved from 74% to 89%, as stated in Table 2.

Table 3 presents a comparison of our proposed hybrid model with state-of-the-art methods on the ISIC 2018 dataset or HAM10000 [50], both of which datasets consist of seven different skin lesion classes. Accuracy is considered an important performance metric for classification tasks, and our model outperformed other state-of-the-art methods. Mporas et al. [51] used traditional machine learning with image preprocessing for hair removal and image segmentation followed by AdaBoost with random forest. Performance didn’t improve much. Milton et al. [52], Majtner et al. [53] and Gessert et al. [54] all employed ensemble models, but there are big differences in their performances. Milton [52] proposed the PNASNet-5-Large model. Chaturvedi et al. [55] utilized a MobileNet model. They were using single models. Therefore, using single models is not necessarily worse than ensemble models. Mobiny et al. [39] boosted the diagnostic performance of the standard DenseNet-169 model from 81.35% to 83.59% by Bayesian deep networks. In our model improvement, we only used a hybrid model and focused on dealing with the problem of data imbalance. Our model accuracy increased from 74.21% to 89.48%. All these experiments have a unified comparison standard, so it can also explain our work is meaningful.

Figure 5 provides an additional representation of the results for more clarity. The left column is the AUC value for each class, and the right column is the average of the seven classes. From the average AUC value on the right, we can see that when using different loss functions, the micro-average ROC outperforms slightly, but the macro-average has a significant improvement. In particular, we used the ViT model, which increased from 0.72 in Figure 5b to 0.84 in Figure 5h with macro-average ROC curve. While Figure 5k presents the AUC of the proposed hybrid model for different skin cancer classes, in which MEL achieves the highest AUC of 93.3 among the six models. Figure 5l reported a mean AUC for all classes, where it reached 0.96 with micro averaging and 0.90 with macro averaging, respectively(an improvement of %7 and 18%). Figure 5 demonstrates that the hybrid model with the FL classifier helps in skin lesion classification with dermoscopic images.

## 5. Conclusions and Future Work

In this paper, there six experiments on the ISIC 2018 dataset were conducted on dermoscopic images. Our results show an improvement in the accuracy of classification results from 74% to 89%. The six different models were proposed to automatically classify skin cancer into seven categories.

The FL approach reduces the impact of imbalanced classes of skin lesions by focusing the loss on minority classes. The results have validated the hybrid model including the FL algorithm which improves the training performance in skin cancer classification.

The severe class imbalance of real-world datasets is still a major point that needs to be addressed. Experimental results indicate that the FL function provides a significant boost to the performance of commonly used CE loss functions for training CNNs on the ISIC 2018 datasets. Our methods offer useful guidelines for researchers working in domains with skin cancer classification.

In future work, the validity of the proposed model can be extended to other databases, such as the ISIC 2019 [57] and ISIC 2020 [58]. In addition, more backbones, such as EfficientNet [59], can be investigated for training models and improving the classification performance. We also consider using Bayesian methods, combined with traditional machine learning methods, to try more techniques to address the data imbalance problem.

## Figures and Tables

**Figure 1 diagnostics-13-00072-f001:**
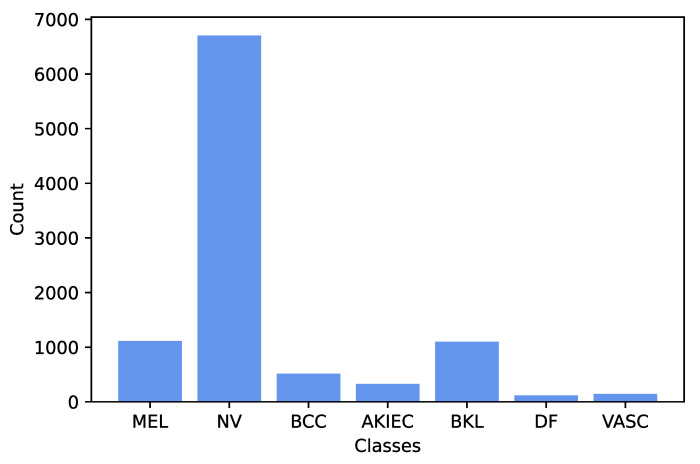
Histogram of the classes in International Skin Imaging Collaboration (ISIC) 2018 dataset. AKIEC = actinic keratosis; BCC = basal cell carcinoma; BKL = benign keratosis lesion; DF = dermatofibroma; MEL = melanoma; NV = melanocytic nevus; VASC = vascular lesion.

**Figure 2 diagnostics-13-00072-f002:**
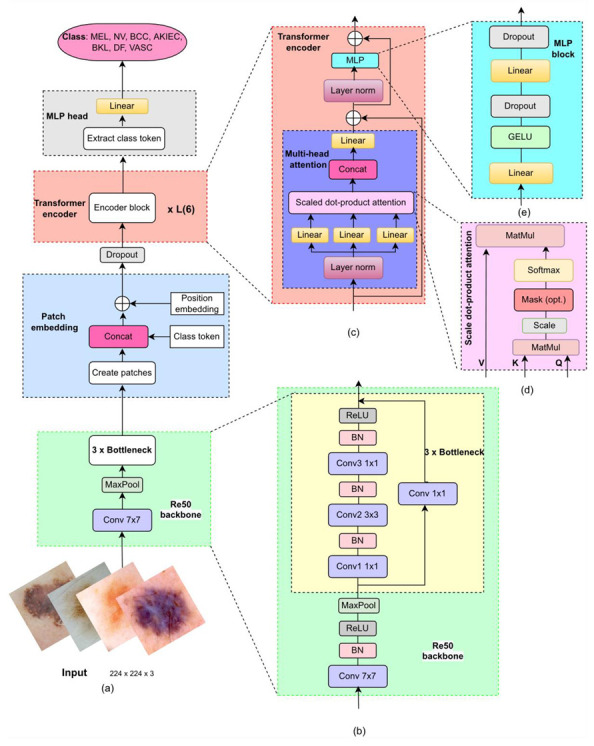
Overview of the proposed hybrid architecture: (**a**) The model comprises two main parts: the convolutional neural network (CNN) and the vision transformer (ViT); (**b**) the CNN for feature map generation; (**c**) the transformer encoder; (**d**) scaled dot product attention; and (**e**) multi-layer perceptron (MLP). The ViT works on global attention encoding for the MLP head classification. GELU = Gaussian error linear unit.

**Figure 3 diagnostics-13-00072-f003:**
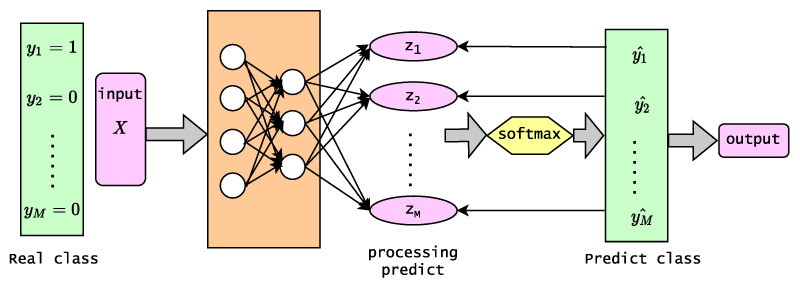
Calculation process of cross-entropy(CE) loss function.

**Figure 4 diagnostics-13-00072-f004:**
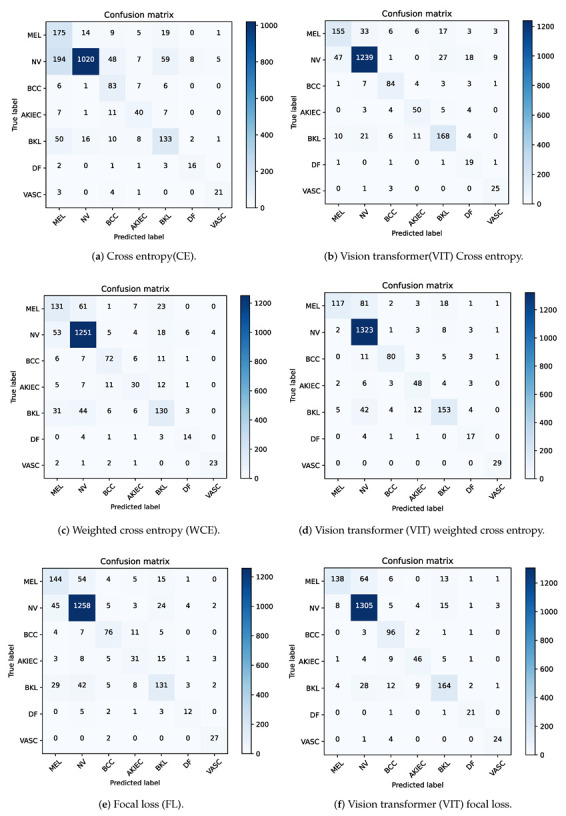
(**a**–**f**) Confusion matrix results of different methods used on the International Skin Imaging Collaboration (ISIC) 2018 dataset. AKIEC = actinic keratosis; BCC = basal cell carcinoma; BKL = benign keratosis lesion; DF = dermatofibroma; MEL = melanoma; NV = melanocytic nevus; VASC = vascular lesion.

**Figure 5 diagnostics-13-00072-f005:**
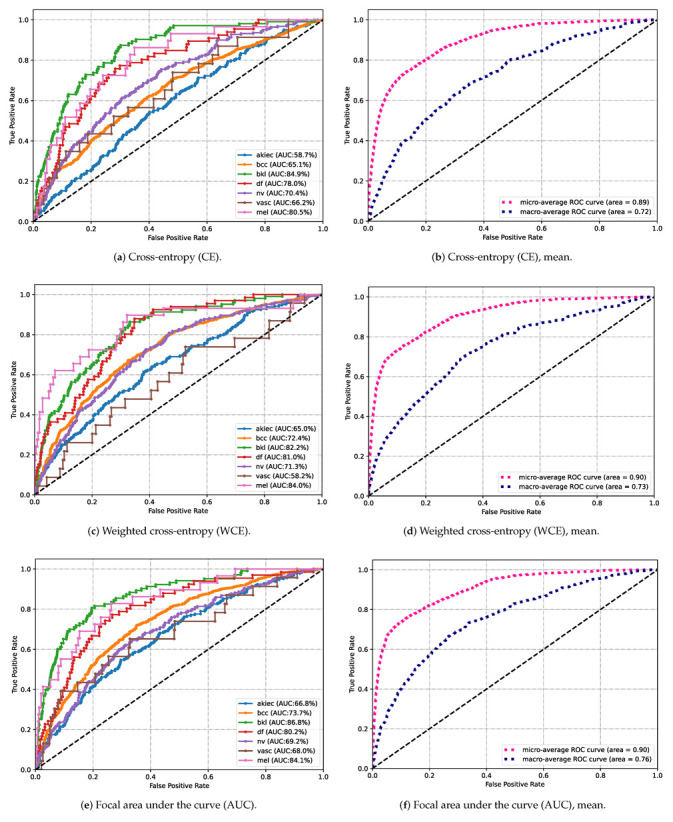
(**a**–**l**) Area under the receiver operating characteristic (ROC) curve (AUC) comparison between the proposed model for classification of the International Skin Imaging Collaboration (ISIC) 2018 dataset. The different classes (AKIEC = actinic keratosis; BCC = basal cell carcinoma; BKL = benign keratosis lesion; DF = dermatofibroma; MEL = melanoma; NV = melanocytic nevus; VASC = vascular lesion) are presented on the left and the mean on the right.

**Table 2 diagnostics-13-00072-t002:** Performance comparison of six proposed models on the International Skin Imaging Collaboration (ISIC) 2018 skin lesion classification dataset using different standard metrics.

Model		Micro Average(%)			Weighted Average(%)			OA *
		Precision	Recall	F1 Score	Precision	Recall	F1 Score	
ResNet-50	CE	62.87	71.17	65.39	82.00	74.21	76.38	74.21± 1.91
	WCE	69.09	66.67	67.75	81.93	82.34	82.09	82.34± 1.67
	FL	69.90	69.14	69.36	83.28	83.74	83.46	83.74± 1.62
Hybrid	CE	70.75	80.63	74.21	87.73	86.78	87.11	86.78± 1.48
	WCE	80.91	77.85	78.20	88.29	88.13	87.37	88.13± 1.42
	FL	82.12	81.53	81.10	89.61	89.48	89.09	89.48± 1.34

CE = cross-entropy; FL = focal loss; OA = overall accuracy; WCE = weighted cross-entropy. * 95% confidence intervals (CI) are included here.

**Table 3 diagnostics-13-00072-t003:** The International Skin Imaging Collaboration (ISIC) 2018 and Human Against Machine with 10,000 training images (HAM10000) datasets. The asterisk (∗) sign on some models shows that auxiliary processing stages and methods were exploited to improve the performance. Our models are given in bold. ANN = artificial neural network; CNN = convolutional neural network; FL = focal loss; ML = machine learning; SVM = Support Vector Machine.

Source	Method	Accuracy (%)
[51]	AdaBoost + random forest	73.08
[51]	RGB+HSV+YIQ color model ∗	74.26
[52]	Ensemble	73.00
[52]	PNASNet ∗	76.00
[56]	ResNet-50 + Forest	80.04
[53]	VGG16 + GoogLeNet Ensemble	81.50
[54]	Densenet121 with SVM	82.20
[54]	Ensemble of CNN ∗	85.10
[55]	MobileNet	83.10
[39]	DenseNet-169	81.35
[39]	Bayesian DenseNet169 ∗	83.59
[20]	DL + Random Forest	85.70
**Our baseline**	**ResNet-50**	**74.21**
**Our best model**	**Hybrid+FL**	**89.48**

## Data Availability

We analyzed the public dataset in this study. It is available at https://challenge.isic-archive.com/data/, accessed on 15 July 2022.

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
