# Peer review of "A Deep CNN Transformer Hybrid Model for Skin Lesion Classification of Dermoscopic Images Using Focal Loss"

_diagnostics, 2022, doi:10.3390/diagnostics13010072_

Round 1

Reviewer 1 Report

In this study, the authors used an "end-to-end CNN transformer hybrid model with a focal loss" to detect skin lesions. Despite the fact that the study is well presented, detailed, and well-written, I do not believe it should be published. This is due to the lack of novelty and the lack of advantages as compared to similar studies. Consider, for instance:

1- "To the best of our knowledge, this is the first network to introduce Transformers into CNN and improve the classification of skin lesions."

I find this statement to be strange. Because. Many studies use transformers for skin lesion classification, even for the ISIC2018 dataset.

2- The comparisons selected in Table 2 are outdated and very selective. It is easy to find a lot of studies with much better performance by conducting a quick search.

Reviewer 2 Report

Nie et al. introduce transformers into CNN and demonstrate improved diagnostic performance on classifying skin lesions. The writing is clear and comprehensive. The presentations are professional so I would like to endorse acceptance of the manuscript after the points addressed.

1. As a paper addressing clinical issue, I am expecting metrics that are commonly used in clinical community. Thus, sensitivity and specificity for detecting malignant diseases (eg. melanoma,basal cell carcinoma) are suggested to be added in the manuscript.

2. Validation details are missing in the manuscript. Would the superior performance of the proposed method possibly be caused by biased validation method? Readers need more related information to be convinced.

3. On reporting the metrics of performance, I would like to suggest reporting mean as well as 95% CI. When comparing the performance, statistical tests are also necessary to tell which model attains higher performance.

4. As a diagnostic model with superior performance, do you consider prepare a Github link with Colab so that colleagues in the community can upload their picture to get a prediction by the proposed model?

Reviewer 3 Report

The authors propose a hybrid model based on CNN, Transformers, and MLP for dermoscopic image classification. The manuscript is clear and comprehensive. However, there are several improvements that could be made in this manuscript that prevent it from being published in its current form.

  - The related work section is presented along with the introduction section. In my opinion, it is interesting to create a new section for this purpose.   - On the basis of the preceding, related works are scarce. Besides expanding the literature research in the area, it is also worth mentioning work that deals with images in similar contexts, such as https://www.ijimai.org/journal/bibcite/reference/3204 and https://www.ijimai.org/journal/bibcite/reference/3037   - Additionally, also related with the literature, I think it is important to add a table at the end of the section providing a summarized description of the different approaches and work studied, highlighting the main differences from the authors' approach.   - The authors are aware of the class imbalance of the dataset, also they use specific accuracy metrics because of this. Have the authors considered the possibility of downsampling or upsampling the classes to balance the dataset?   - I know that data augmentation techniques have been applied to increase the dataset instances and balance the dataset. Have the author considered applying other techniques to generate synthetic data?   - During the experimentation, I miss a comparative of the performance of the authors approach with other paper analyzed or with other techniques in or out this field. Would it be possible?   - Is there any limitation of the work, bias on data, or thread to validity worth discussing after the discussion of the experimental results?  

- All the material used in the experimentation should be further discussed and provide the means to facilitate the experiments' reproducibility.

Round 2

Reviewer 3 Report

Accept as is